# Selective Antibody-Free Sensing Membranes for Picogram Tetracycline Detection

**DOI:** 10.3390/bios13010071

**Published:** 2022-12-31

**Authors:** Hamdi Ben Halima, Abdoullatif Baraket, Clara Vinas, Nadia Zine, Joan Bausells, Nicole Jaffrezic-Renault, Francesc Teixidor, Abdelhamid Errachid

**Affiliations:** 1Institut de Sciences Analytiques (ISA)-UMR 5280, Université Claude Bernard Lyon 1, 5 Rue de la Doua, 69100 Lyon, France; 2Inorganic Materials Laboratory, Institut de Ciencia de Materials de Barcelona (ICMAB-CSIC), Campus de la UAB, Bellaterra, 08193 Barcelona, Spain; 3Institute of Microelectronics of Barcelona (IMB-CNM, CSIC), Campus UAB, Bellaterra, 08193 Barcelona, Spain

**Keywords:** tetracycline, electrochemical impedance spectroscopy, single-walled carbon nanotubes, tetracycline/[*o*-COSAN]^−^ ion-pair complex, polyvinyl chloride, selectivity

## Abstract

As an antibody-free sensing membrane for the detection of the antibiotic tetracycline (TC), a liquid PVC membrane doped with the ion-pair tetracycline/θ-shaped anion [3,3′-Co(1,2-C_2_B_9_H_11_)_2_]^−^ ([*o*-COSAN]^−^) was formulated and deposited on a SWCNT modified gold microelectrode. The chosen transduction technique was electrochemical impedance spectroscopy (EIS). The PVC membrane was composed of: the tetracycline/[*o*-COSAN]^−^ ion-pair, a plasticizer. A detection limit of 0.3 pg/L was obtained with this membrane, using bis(2-ethylhexyl) sebacate as a plasticizer. The sensitivity of detection of tetracycline was five times higher than that of oxytetracycline and of terramycin, and 22 times higher than that of demeclocycline. A shelf-life of the prepared sensor was more than six months and was used for detection in spiked honey samples. These results open the way to having continuous monitoring sensors with a high detection capacity, are easy to clean, avoid the use of antibodies, and produce a direct measurement.

## 1. Introduction

Sensors are devices that detect and respond to some type of input from the physical environment. They are found everywhere in our daily lives and the aim is to make them practical and easy to handle, moving away from expensive complex instrumentation to small inexpensive systems that can be operated by anyone. In the chemical sensing field, potentiometry is the archetype of simplicity: simple sensing material commonly made of a membrane with standard and low-weight electronic equipment. The sensing material is responsible for interacting with the analyte, producing a change in the interfacial potential that is then transformed into a readable signal by the transducer [1,2,3,4,5,6,7,8,9,10,11,12,13]. For the popular ion-selective electrodes (ISE), the sensing membrane consists of a polymer matrix usually based on plasticized poly(vinyl chloride) and one electroactive additive, commonly a lipophilic salt [4,14,15]. The selectivity of these membranes has been related to the Hofmeister series, however when a such relationship is not sought, the membrane composition is complemented with a second electroactive additive or ionophore, usually a complexing selective ligand, e.g., valinomycin for K^+^. Despite many ongoing theories about the mechanism of charge transfer at the interfaces, the reality is that the reasons behind the sensing and the selectivity are still not well inferred. Far less complex are the biosensors, for which the selectivity to the analyte is conceptually well interpreted because of the specific complementarity between enzyme or antibody, or aptamer and the analyte. It is generally accepted that an interaction between the analyte and the sensing material is a necessary condition for the feasibility of any chemical sensor. Non-covalent interactions may lead to a reversible or partially reversible response. Conversely, covalent bonding, which can provide high selectivity and sensitivity, often leads to an irreversible response. On the other hand, potentiometry, in practical terms, shows low detection limits near 10^−6^ or 10^−7^ M, whereas electrochemical antibodies or aptamer biosensors can go down to the pg [1,3,16]. As they are based on biological material, biosensors have associated problems such as the influence of the environment, temperature, and atmosphere, among others, which make them especially delicate and require rigorous storage conditions.

A few years ago we introduced, in potentiometry [1,3,5,17], the θ-shaped molecule [3,3′-Co(1,2-C_2_B_9_H_11_)_2_]^−^ ([*o*-COSAN]^−^) into PVC membranes made of PVC: plasticizer in a ratio of 1:2. The principal motivation was that [NRR’R’’R’’’][3,3′-Co(1,2-C_2_B_9_H_11_)_2_] salts are highly insoluble in water, whereas they are highly soluble in many organic solvents. At that time, we did not know much more about the physicochemical characteristics of the θ-shaped molecule [3,3′-Co(1,2-C_2_B_9_H_11_)_2_]^−^. The membrane did not contain any other components besides the cation of the analyte and the plasticizer. The potentiometry results on the macroscopic electrodes were excellent, with the membrane set on a solid support made of graphite powder mixed with Epoxy Resin, providing very good selectivity; whereas the low limit of detection (LOD) remained in most cases about 10^−6^ M. Electrodes for antibiotics [18], amino acids [19] and other biomolecules [20,21] were developed. The most surprising aspect was that such a simple membrane composition was able to discriminate a chiral amino acid from its enantiomer, with selectivity coefficients Kxy between 10^−2^ and 10^−3^ [22].

Since our first publications in the area of potentiometry with [NRR′R″R‴][3,3′-Co(1,2-C_2_B_9_H_11_)_2_], we and others have contributed very much to the understanding of the physicochemical properties of [Co(C_2_B_9_H_11_)_2_]^−^. With regards to its sensing ability, we would construe that its tunable reversible redox potential [23,24,25,26], its self-assembling capacity in solid and aqueous solution [27,28], its capacity to produce hydrogen and dihydrogen bonds [29,30,31], its ability to dope conducting organic polymers [26,32], and its amphiphilic character [33], among others, are key points to explain the extraordinary performance of this anion. However, as stated above, it was not possible to overcome the Low LOD. We wondered if this problem could be solved with another electrochemical transducer, while keeping the same membrane composition so that this sort of electrochemical measurement could approach the biosensors in terms of low LOD. Hopefully, this would lead to the sensing materials being able to amalgamate the good points of potentiometry with the low LOD, and the selectivity of biosensors. We sought to use the electrochemical impedance spectroscopy (EIS) technique, and we decided to initiate our works with tetracycline. The aim was to discover if the impedance increased with increasing concentrations of analyte, and whether there was any selectivity with structurally closely related chemicals. For example, the set of tetracycline (TC) antibiotics, that would show the necessity for the analyte inside the membrane, whether there was a relationship between signal and concentration, and whether there was any real hope of getting a low value of the low detection limit.

As we shall see in this paper, the results are highly encouraging. We have made an ISE PVC membrane incorporating protonated tetracycline compensated by [3,3′-Co(1,2-C_2_B_9_H_11_)_2_]^−^ and, as an extra component, carbon nanotubes as an inner conductivity-enhancing layer. Tetracycline was chosen because there is a set of antibiotics all having the same tetracycline skeleton which, because of their very similar structures, will allow appreciation of the selectivity that can be achieved. Tetracycline is used to treat infections caused by bacteria in the respiratory tract, lymphatic, intestinal, genital and urinary systems, on the skin and eyes and certain other infections that are spread by infected animals [34,35,36]. The molecular structure of tetracycline and of selected similar molecules are shown in Figure 1a.

Based on the literature, there are several analytical methods that have been described for tetracycline monitoring, including: high performance liquid chromatography coupled with mass spectrometry [37], capillary electrophoresis coupled with electrochemiluminescence [38], ELISA (enzyme-linked immune-sorbent assay) [39,40], liquid chromatography-mass spectroscopy (LC-MS) [41], and spectroscopy analysis [42,43]. However, these methods suffer from a lack of sensitivity compared to chromatographic techniques [44]. Commonly, the principal limitations of these steps lie in the lack of sensitivity, high cost, time-consuming implementation and the requirement for sophisticated technical skills [45]. New solutions need rapid, simple and accurate methods for the on-site screening of low TC residues without any supplementary steps such as extraction or clean-up. Owing to their advantages of high selectivity and rapid detection, several optical and electrochemical biosensors have been investigated [46]. For this reason, aptamer-based sensing techniques were widely used for the food safety determination. Above all, there is a growing rise in aptasensor fabrication for TC detection [47,48], with only some applications to honey samples [49,50]. Unfortunately, the mean disadvantage of these systems is their relatively low detection signals [51]. In addition, in some recent works, a molecularly imprinted sensor had been successfully applied to the analysis of antibiotic residues in honey samples [52,53]. However, these types of sensors suffer from reversibility and a short shelf-life when the imprinted membrane is fragile. 

Tetracycline and tetracycline antibiotics (shown in Figure 1) all have a common linear fused tetracyclic nucleus that differ with the functional groups attached. All of them contain only one protonable amine group N(Me)_2_. Therefore, the salt [tetracycline-H][3,3′-Co(1,2-C_2_B_9_H_11_)_2_] was expected. It was thus anticipated that PVC/plasticizer/carbon nanotubes/[tetracycline-H][3,3′-Co(1,2-C_2_B_9_H_11_)_2_] would be an excellent candidate to test the feasibility of the electrochemical sensor based on an EIS transducer. Indeed, this proved to be the case with a lower detection limit with excellent selectivity.

## 2. Materials and Methods

### 2.1. Materials and Chemicals

Tetracycline (TC), N-hydroxysuccinimide (NHS), 11-amino-1-undecanethiol, phosphate buffer solution (PBS) tablets, 1-ethyl-3-(3-dimethylaminopropyl) carbodiimide (EDC), Ethanol, sodium dodecyl sulfate (SDS), Polyvinyl chloride (PVC), o-nitro phenyl octyl ether (NPOE), di-octyl phthalate (DOP), dibutyl phthalate (DBP), dibutyl sebacate (DBS), hydrochloric acid (HCl), diethyl ether, tetrahydrofuran (THF), di-octyl phthalate, bis(2-ethyl hexyl) sebacate were purchased from Sigma-Aldrich (France). The standard solutions and buffers were prepared with Millipore Milli-Q nanopure water (resistivity > 18 MW cm) which is produced by a Millipore Reagent Water System (France). Epoxy resin EPO TEK H70E 2LC was from Epoxy Technology, France. Cs[*o*-COSAN], was synthesized from 1,2-*closo*-C_2_B_10_H_12_ from Katchem Spol.sr.o (Kralupy nad Vltavou, Czech Republic), as reported in the literature [54]. The Na[*o*-COSAN] was obtained by means of cationic exchange resin from Cs[*o*-COSAN] following the previously described procedure [31].

### 2.2. Preparation of the Ion-Pair Complex [tetracycline-H][Co(C_2_B_9_H_11_)_2_]

Tetracycline hydrochloride (40 mg, 0.083 mmol) was dissolved in diluted hydrochloric acid (~25 mL). After agitating and obtaining a clear solution, Na [3,3′-Co(1,2-C_2_B_9_H_11_)_2_] (0.083 mmol) in 10 mL of diluted hydrochloric acid (1 or 3 M) were added. Almost instantly a precipitate appeared. The mixture was stirred for 5 min and left to rest for an additional 15 min. The orange solid was filtered through a Buchner funnel with filter paper. After rinsing first with 10 mL diluted hydrochloric acid (0.1 M) and then 2 × 10 mL of deionized water, the filter paper was carefully removed and placed in a round bottom flask with a ground glass joint for active 0.1–0.01 mm vacuum at room temperature. After 4–5 h, the solid was collected and was ready for the membrane preparation. The formula of the ion-pair complex is shown in Figure 2.

### 2.3. Preparation of the Gold Microelectrode Modified by Single-Walled Carbon Nanotubes and Electrochemical Measurements

The microelectronics fabrication process for the microelectrodes was performed at Centro Nacional de Microelectronica (CNM). The process started with a thermal oxidation process to grow a thick oxide layer (8000 Å) on 100 mm diameter P-type <100> silicon wafers with a nominal thickness of 525 mm. The working microelectrode was made with a metal layer consisting of a thin titanium film (10 nm) promoting gold adhesion plus 250 nm of gold. After that, photoresist layer was spin-coated with a spinning speed of 3000 rpm and was exposed in UV light with a pattern mask. Etching away the exposed photoresist was performed with the developer OPD4262 from Fujifilm. The remaining photoresist corresponded exactly to the microelectrodes, and then the gold, unprotected by the photoresist, was etched away. The next step consisted of the deposition of two PECVD (Plasma-Enhanced Chemical Vapor Deposition) layers of SiO_2_ (4000 Å) and Si_3_N_4_ (4000 Å), to act as a passivation layer. The second photolithographic process was performed to open the passivation on the active Au microelectrodes (300 µm × 300 µm; area: 9 · 10^−4^ cm^2^) and on the soldering pads. The structure of the microelectrodes is shown schematically in Figure 3. Wire bonding was performed using a Kulicke & Soffa 4523 A Digital instrument from Kulicke & Soffa, Singapore.

Microelectrodes were exposed to UV using UV/ozone Procleaner^TM^ (BioForce, Germany) for 30 min for cleaning and activation by creating –OH groups. These were modified by adsorption of 11-amino-1-undecanthiol-HCl (aminothiol), by dipping the electrode in a 10 mM solution of the aminothiol in ethanol (EtOH) for one night at 4 °C. Then, the microelectrode was rinsed with ethanol to remove unbound thiols and dried with N_2_. In a separate beaker, (0.17 g/L, 0.25 mL) of COOH-SWCNT (Carboxylic functionalized single-walled carbon nanotubes) were added to an aqueous solution of SDS (sodium dodecyl sulfate, 0.1 M), 0.1 M in NHS (N-hydroxysuccinimide) and 0.4 M in EDC (1-ethyl-3-(3-dimethylaminopropyl) carbodiimide-HCl), and the reagents were left in contact for one hour. Then, the microelectrode functionalized with aminothiol was placed with the solution containing the activated SWCNTs-COOH for 2 h at room temperature. Finally, the SWCNTs functionalized microelectrode was rinsed with ultrapure water to remove the unbounded SWCNTs and dried with N_2_.

All electrochemical measurements were carried out in a Faraday cage at room temperature (r.t., 22 ± 2 °C). The electrochemical experiments were carried out using a VMP3 multichannel potentiostat (Biologic-EC-Lab, Seyssinet-Pariset, France). Data were acquired and analyzed using EC-Lab software V11.30. The EIS data were fitted by using the Randomize + Simplex method.

### 2.4. Liquid Membrane Preparation

In earlier potentiometric works related to [3,3′-Co(1,2-C_2_B_9_H_11_)_2_]^−^, it was found that the most suitable mix of electroactive material, plasticizer, and PVC powder for membranes was either 3% or 7% of the electroactive material, 63% of plasticizer and 34% or 30% of PVC powder, all in weight percentages that were dissolved in THF. The membrane solution was prepared as follows: 43 mg of PVC were dissolved by stirring in 1.5 mL of tetrahydrofuran (THF) until a viscous but clear solution was obtained. Then, 10 mg of [tetracycline-H]/[3,3′-Co(1,2-C_2_B_9_H_11_)_2_] and 90 mg of plasticizer were added. The resulting dispersing solution is deposited on the surface of the electrode body. In this paper, the results of three membranes (membranes named 1,2,3) are reported; the differences between the three membranes lie in the concentration of HCl in the preparation of the ion-pair complex (1 or 3 M) and in the plasticizer used: *Membrane 1*, HCl 1 M, 91.75 µL di-octylphthalate; *Membrane 2*, HCl 3 M, 91.75 µL di-octylphthalate, *Membrane 3*, HCl 3 M, 98.5 µL bis (2-ethyl hexyl) sebacate.

The membrane was drop-cast (2 µL drop) onto the gold microelectrode already functionalized with the SWCNTs layer, and the solvent was allowed to evaporate at an ambient temperature for 24 h [1,5]. Once the membrane had dried, the microelectrode was then immersed for 24 h at 4 °C in tetracycline at 10^−3^ M in order to achieve appropriate conditioning of the PVC tetracycline membrane. The microelectrodes were stored at room temperature for future use.

## 3. Results and Discussion

### 3.1. Characterization of the [o-COSAN]^−^/Tetracycline Ion-Pair Complex

In order to characterize our ion-pair complex, different techniques were used such as Proton Nuclear Magnetic Resonance (^1^H NMR), Carbon Nuclear Magnetic Resonance (^13^C{^1^H} NMR), Fourier Transform Infrared Spectroscopy (FTIR), Matrix-Assisted Laser Desorption/Ionization-Time-of-Flight Mass Spectrometry (MALDI-TOF-MS) and Elemental Analysis. 

The ratio [tetracycline-H]/[3,3′-Co(1,2-C_2_B_9_H_11_)_2_] (Figure 1b) is calculated from ^1^H-NMR, in d_6_-acetone, given the areas of the unquestionable key hydrogen atoms of the cation and the anion. The C-*H* resonances at the aromatic region of the spectrum (close to 7 ppm) correspond to the aromatic hydrogen atoms of the tetracycline cation, and the signal that appears close to 4 ppm corresponds to the four hydrogen atoms (C_cluster_-*H*) of the anionic [3,3′-Co(1,2-C_2_B_9_H_11_)_2_] cluster. Additionally, the ^1^H NMR spectrum displays the other signals corresponding to [*o*-COSAN]^−^ and tetracycline that unambigously indicates the presence of the [*o*-COSAN]- anion, and the protonated tetracycline cation in the ion-pair complex. By integration, the ratio [tetracycline-H]/[*o*-COSAN]^−^ in the ion-pair complex was calculated (Appendix A); there are three aromatic protons in each protonated tetracycline cation, and its integration area is around 3.5 per the 4 area corresponding to each [*o*-COSAN]^−^ anion, which indicates that there is one molecule of [*o*-COSAN]^−^ per one molecule of protonated tetracycline.

The ^13^C{^1^H} NMR spectrum (Appendix A) displays the signals of the different carbons of the protonated tetracycine molecules, which are represented with letters, as well as the signal of C_cluster_-H vertices of the [*o*-COSAN]^−^ that appear at 51.0 ppm. The signals without letters correspond to the solvent (deuterated acetone and other solvents used in the preparation).

On the other hand, to observe if in the sample there is [*o*-COSAN]^−^, a signal should appear in the range 2520–2550 cm^−1^ in the IR spectrum (Appendix A).

FTIR spectrum of our ion-pair complex displays the ν(O-H) stretching vibration at 3611 cm^−1^; ν(N-H; O-H) at 3370–3611 cm^−1^; ν(C_aryl_-H, C_cluster_-H) at 3047 cm^−1^; ν(B-H) at 2539 cm^−1^; ν(C=O) at 1659 cm^−1^; ν(C=O_amide_; C=C; C=C_aromatic_; N-H) at 1529–1612 cm^−1^; ν(C-H) at 1322–1452 cm^−1^; ν(C-O; C-C; C-N) at 1097–1322 cm^−1^; ν(C-H; C=C) at 721–983 cm^−1^. 

In MALDI-TOF-MS we can see the mass spectrum of the negative and positive part of the sample, and know the molecular mass of both species. In Appendix A we can see the positive part of the mass spectrum of our prepared ion-pair complexes; the negative part is the signal of [*o*-COSAN]^−^ at 324 m/z. These spectra were created with a matrix. The Mass Spectra was recorded on a matrix, therefore not all signals correspond to the mass of the cation.

In addition, Appendix A shows the elemental analysis of our ion-pair complex, which contains the theoretical values of %C, %H, %N and %S and the real values. The real values are very similar to the theory, so we can confirm that our sample has the correct structure.

### 3.2. Electrochemical and Physical Characterization of the Gold Microelectrode

#### 3.2.1. Characterization of the Gold Surface Before and After Immobilization of Amino Thiol Using Cyclic Voltammetry

The gold electrodes were characterized by cyclic voltammetry (CV) before and after the immobilization of the amino thiol. The CV scan was carried out from −0.4 to +0.6 V, with a scanning rate of 80 mV/s in 5 mL of PBS in the presence of the redox couple [Fe(CN)_6_]^3−/4−^. Figure 3 shows that the oxido/reduction peaks of bare gold disappear after the immobilization of aminothiol. The disappearance of the oxido/reduction peaks was caused by the blocking of the aminothiol layer of the gold surface, thus creating a low electron transfer rate. 

#### 3.2.2. EIS Characterization of the Gold Microelectrode at the Different Stages of Modification

The electrode performance was studied by EIS by applying a potential of −0.4 V in a frequency range of 100 kHz to 6 Hz at each individual step of preparation: 1) to the bare Au electrode, 2) after functionalization with the aminothiol, and 3) after grafting of carbon nanotubes. Figure 4 shows the evolution of the conductance of the Nyquist diagrams at the different microelectrode functionalization stages. As expected, the most conductive is the bare electrode, whose conductivity decreases upon the addition of the aminothiol with its long carbon chain (C11). This conductivity is slightly increased by the grafting of carbon nanotubes.

#### 3.2.3. SEM Characterization

Scanning Electron Microscopy (SEM) was employed to investigate the surface morphology of the SWCNTs, fixed on to the gold microelectrode using FEI Quanta FEG 250. Figure 5 confirms the grafting of SWCNTs onto the gold microelectrode functionalized with amine thiol-SWCNTs. The immobilization of SWCNTs onto the microelectrode was confirmed by the small white rods, however, the black spots in the image indicate some spots not covered by SWCNTs.

#### 3.2.4. Characterization by Fourier-Transform Infrared Spectroscopy (FTIR)

The presence of different compounds of our membrane, such as polymer matrix, plasticizer, and [*o*-COSAN]^−^/tetracycline ion-pair complex, were confirmed after deposition onto the microelectrode. First, the FTIR spectrum of membrane 3 after deposition onto the microelectrode was carried out. In parallel, three different reference FTIR spectra of PVC, bis (2-ethylhexyl) sebacate and [*o*-COSAN]^−^/tetracycline ion-pair complex were performed.

FTIR spectra of the different compounds of our membrane: bis (2-ethylhexyl) sebecate (plasticizer) (Appendix A), PVC polymer matrix (Appendix A), and ionophore of cobalt bis(dicarbollide) (Appendix A) were superimposed with the FTIR spectrum of membrane 3 after deposition onto the microelectrode (Figure 6).

Figure 6 confirms the presence of the different compounds of the membrane, since the absorption peaks are localized at the characteristic wavelengths of the membrane immobilized on the microelectrode. These results confirm the success of the immobilization of the membrane onto the microelectrode.

### 3.3. Detection of Tetracycline by Electrochemical Impedance Spectroscopy (EIS)

Impedance was measured using an electrochemical cell consisting of three electrodes (Figure 4). The reference electrode was a saturated calomel electrode, the auxiliary electrode was a platinum wire of 1 mm in diameter and the working electrode was the microelectrode described above. The measurements of tetracycline were carried out in a PBS buffer solution (pH = 7.4), varying the tetracycline concentration from 1 pg/L to 5 ng/L. All experiments were repeated three times to confirm the reproducibility of our biosensor. The experiments were performed in darkness to protect tetracycline, which is a fluorescent compound from degradation, and in a Faraday box to eliminate electrical interference. The Nyquist plots (-Im (Z)/Re (Z)) obtained with the three membranes were recorded while increasing the concentration of tetracycline. The reported results are the mean of the three sensing areas for one electrode (Figure 2).

From the Nyquist diagrams presented in Figure 7, it is noticeable that there is a marked response from the picogram quantities of tetracycline, as can be seen from the separation in the Nyquist plot between 0 pg and 1 pg. There is a steady increase of both Re(Z) and –Im(Z) in parallel with an increasing amount of analyte. In addition, we have noticed that the HCl concentration influences the preparation of [*o*-COSAN]^−^/tetracycline ion-pair complex and of the used plasticizer.

Data fitting on EIS spectra was achieved by using the Randomize + Simplex method using Randles equivalent circuit model [R_s_ + Q_2_/ (R_ct_ + Z_W_)], in which R_s_ corresponds to the resistance of the electrolyte solution; Q_2_ is the phase constant element that is in parallel with R_ct_, which is the charge transfer resistance; and Z_W_ represents the Warburg impedance (Appendix A). The standardization plots were obtained by drawing tetracycline concentrations vs. R_ct_ normalized data (∆R/R= (R_sample_ − R_Ref_)/R_Ref_). The example of the calibration curve of the sensor with membrane 3 is presented in Appendix A. The reproducibility is 6%. 

The sensitivity of the tetracycline sensor is the slope of the straight line. With membrane 3, the sensitivity is 1.83 ± 0.4 ng^−1^·L, as presented in Figure 8. The RSD of 22% is the repeatability obtained between 4 sensors. The sensitivity of the tetracycline sensor with membrane 1 is 0.5 ± 0.2 ng^−1^·L and that of the sensor with membrane 2 is 0.5 ± 0.08 ng^−1^·L, showing the high influence of the used plasticizer and the negligible influence of the HCl concentration in the preparation of the [*o*-COSAN]^−^/tetracycline ion-pair complex.

### 3.4. Selectivity of the Tetracycline Sensors

The selectivity of the tetracycline sensors was obtained by testing oxytetracycline, terramycin, and demeclocycline dissolved in a solution of PBS (pH = 7.4) in quantities ranging from 1 pg/L to 5 ng/L. The electrodes and the setup used were the same as those used for tetracycline, and the impedance of the system were measured. All experiments were done thrice to confirm the reproducibility of the sensor.

The slope of each curve of the three membranes for the detection of tetracycline, oxytetracycline, demeclocycline, and terramycin are drawn as bars in Figure 8. The sensitivity of detection of tetracycline was 5 times higher than that of oxytetracycline and of terramycin, and 22 times higher than that of demeclocycline.

Compared to the published tetracycline biosensors presented in Table 1, our tetracycline sensor presents a detection limit in the lower range. Most presented biosensors need aptamers/Ab TC and antibodies/Ag-TC, which are expensive compared to our developed approach. 

In addition, the use of antibodies and antigen requires storage at 4 °C of their sensor; however, with our developed approach, our sensors could be stored at room temperature. Due to the reversibility of the interactions, there is a limit to the reusability of the tetracycline sensors. The shelf-life of the prepared sensor with membrane 3 was more than six months. For tetracycline biosensors based on antibody, aptamer or MIP, the resusability is quite limited. In Benvidi et al. [61], only five regenerations were possible without any loss of sensitivity.

The proposed mechanism is based on the exchange process that occurs at the membrane interface with the analyte, and is a consequence of the geometry and chemical composition of the cobalt bis(dicarbollide) [3,3′-Co(1,2-C_2_B_9_H_11_)_2_]^−^, particularly the existence of B-H and C_cluster_-H bonds that can generate hydrogen and dihydrogen bonds. We believe this is essential. In their absence, common ion exchangers will not generate strong interactions with either the ammonium cation or with the plasticizer solvent. Therefore, they have high mobility in the membrane. This is not the case with [3,3′-Co(1,2-C_2_B_9_H_11_)_2_]^−^ which does make these strong non-bonding interactions and generates more reticulate, and therefore more stability and a precise concentration of the analyte in the membrane. The cobalt bis(dicarbollide) has dimensions of 1.1 nm in length and 0.6 nm in width and is surrounded by hydrogens, the mentioned B-H bonds, that have considerable hydride character but not enough to be unstable in protic solvents. This sufficient hydride character of the B-H groups enables it to interact strongly with H-N units. The non-bonding interactions B-H···H-N are weak, but if there are many, as is the case, they become a strong interaction. This leads to a very stable concentration of the target analyte within the membrane, and allows for an adequate ion exchange thanks to the ion exchanger capacity of [3,3′-Co(1,2-C_2_B_9_H_11_)_2_]^−^, overall giving the appropriate stability and sensitivity.

The higher sensitivity for tetracycline compared to the similar molecules should be explained by a higher stability of the [*o*-COSAN]^−^/tetracycline ion-pair complex, whereas the similar molecules present a primary amine group which is a prerequisite, apart from terramycin. Demeclocycline presents a chloride group and oxytetracycline, a supplementary OH group which can limit the ion-pair complex stability.

### 3.5. Determination of Tetracycline in Spiked Honey Samples with the Tetracycline Sensor (Membrane 3)

Honey solutions were prepared as follows: 1 g of honey was added to 1 mL of ethanol and maintained in an ultrasonic bath for 30 min. In order to reduce the matrix effect, the samples were then diluted 1/10 (v/v) with PBS and filtered through a 0.8 mm cellulose acetate filter. The samples were then spiked with standard TC solutions, and measurements were carried out through the standard addition method. Three replicates were made for each the two samples. Table 2 summarizes the tetracycline sensor results for the determination of TC in spiked honey samples. Recoveries of 100% to 107% were found, with an RSD of 6%. This indicates that the repeatability of the tetracycline sensor prepared with membrane 3 is acceptable in practice.

## 4. Conclusion

In this work, we have discussed the development process of a novel sensitive and highly selective tetracycline sensor, based on gold microelectrodes modified by single-walled carbon nanotubes (SWCNTs) and a plasticized PVC membrane doped with a [*o*-COSAN]^−^/tetracycline ion-pair complex. The ion-pair complex was synthesized and characterized using different techniques such as Proton Nuclear Magnetic Resonance (^1^H NMR), Carbon Nuclear Magnetic Resonance (^13^NMR), Fourier-Transform Infrared Spectroscopy (FTIR), Matrix-Assisted Laser Desorption/Ionization- Time-of-Flight Mass Spectrometry (MALDI-TOF MS) and Elemental Analysis. The electrochemical detection was performed by electrochemical impedance spectroscopy (EIS) for three different membranes. Membrane 3 with bis(2-ethyl hexyl) sebacate as the plasticizer, shows the highest sensitivity compared to the other membranes. The high selectivity of our prepared tetracycline sensors was demonstrated by analyzing solutions containing similar molecules, namely oxytetracycline, demeclocycline, and terramycin. The proposed approach presents a high sensitivity compared to published tetracycline biosensors; moreover, it is a cheap solution compared to those requiring aptamers or antibodies, and it is a durable solution because of the reversibility of the interactions with the [*o*-COSAN]^−^/tetracycline ion-pair complex, and the shelf-life time of the prepared sensor was found to be more than six months. The work carried out in this paper aims at providing a solution to the food or drug control of improving portable instrumentations while incorporating new technologies.

## Figures and Tables

**Figure 1 biosensors-13-00071-f001:**
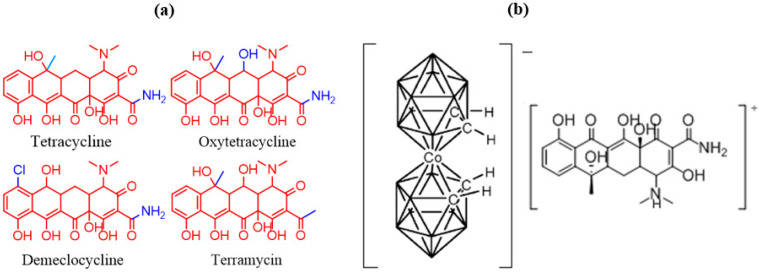
(**a**) Tetracycline and the chosen similar molecules and molecular structure of [*o*-COSAN]^−^/tetracycline ion-pair complex (**b**).

**Figure 2 biosensors-13-00071-f002:**
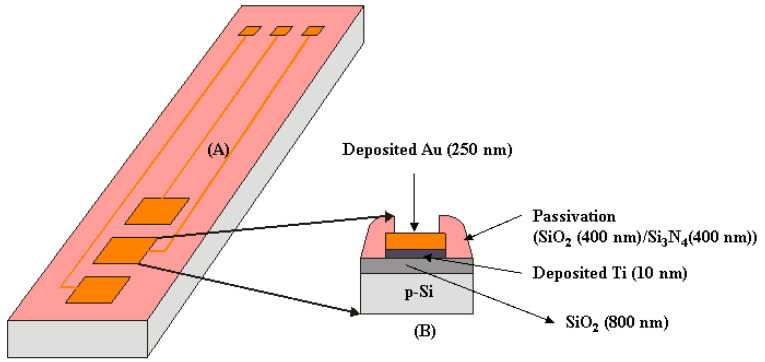
(**A**) Schematic view of microelectrodes based on silicon technology; (**B**) Cross section of one planar microelectrode.

**Figure 3 biosensors-13-00071-f003:**
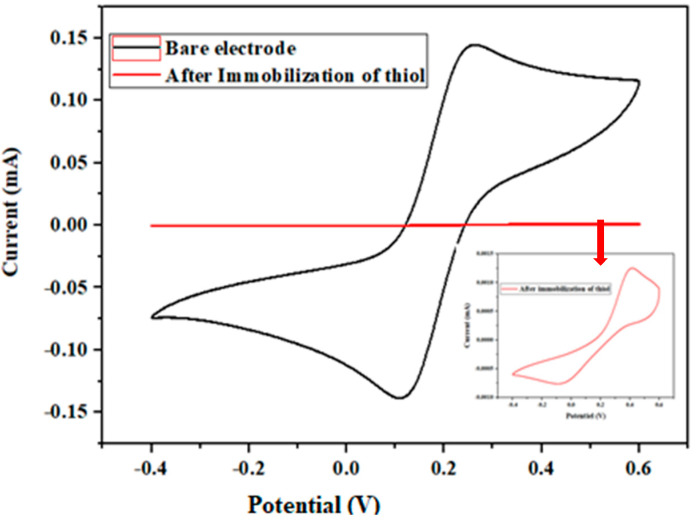
Characterization of gold microelectrode by cyclic voltammetry (CV), before and after immobilization of aminothiol in 5 mM [Fe(CN)_6_]^3−/4−^ and PBS, from −0.4 to +0.6 V at a scan rate of 80 mv/s. Insert: Enlarged cyclic voltammetry (CV) after immobilization of aminothiol.

**Figure 4 biosensors-13-00071-f004:**
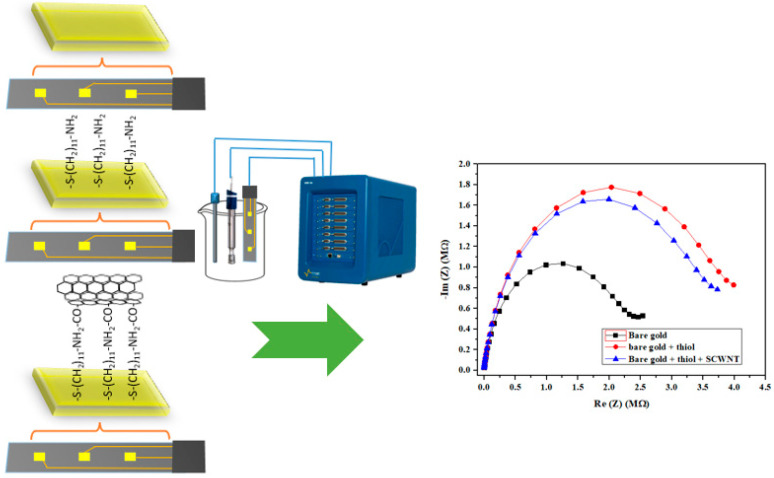
Nyquist diagram (-Im (Z) vs. Re (Z)) corresponding to the impedance measurements by applying a potential of −0.4 V and amplitude potential of 10 mV between a frequency range of 100 kHz to 6 Hz for the various grafted layers on the gold microelectrode. Bare gold (black), after the immobilization of the aminothiol (red) and after grafting SWCNTs (blue).

**Figure 5 biosensors-13-00071-f005:**
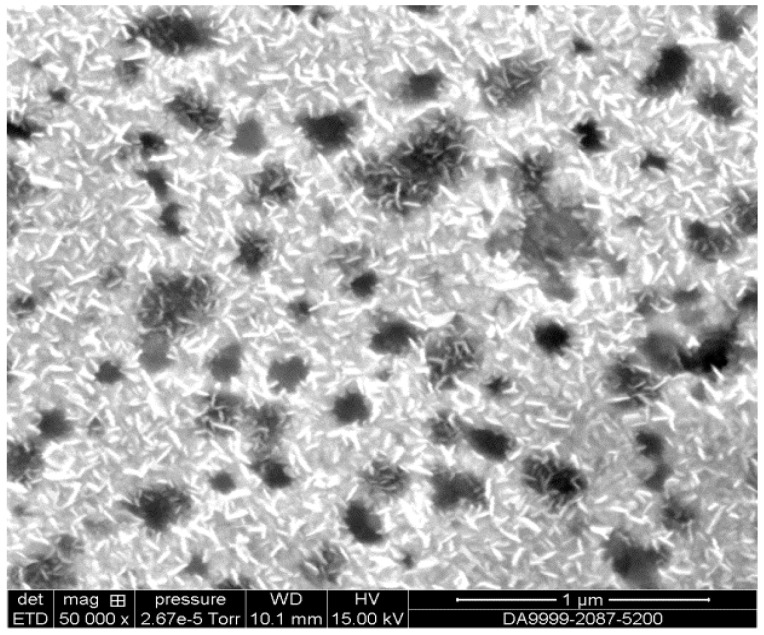
SEM image under functional microelectrode (amine thiol + SWCNTs).

**Figure 6 biosensors-13-00071-f006:**
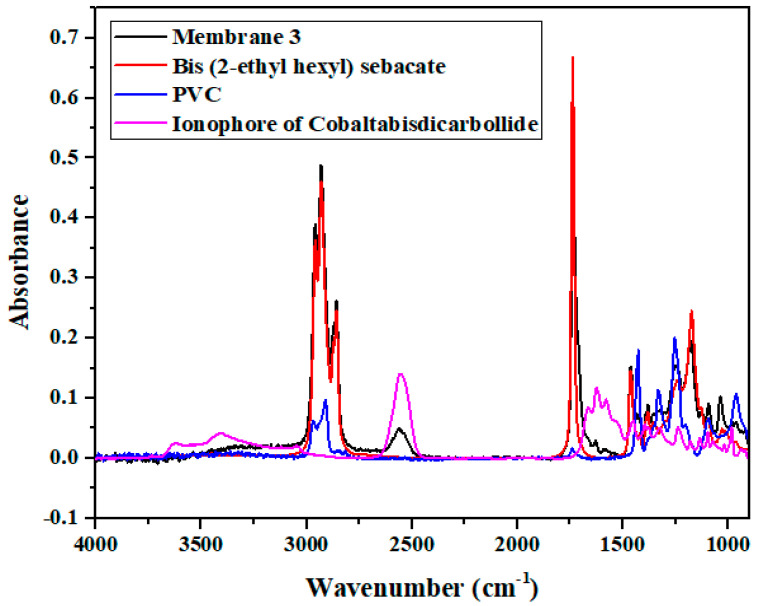
FTIR Spectrum of membrane 3 after deposition onto the microelectrode (black), bis (2-ehylhexyl) sebacate (red), PVC (blue), Ionophore of Cobalt bis(dicarbollide)–Tetracycline (pink).

**Figure 7 biosensors-13-00071-f007:**
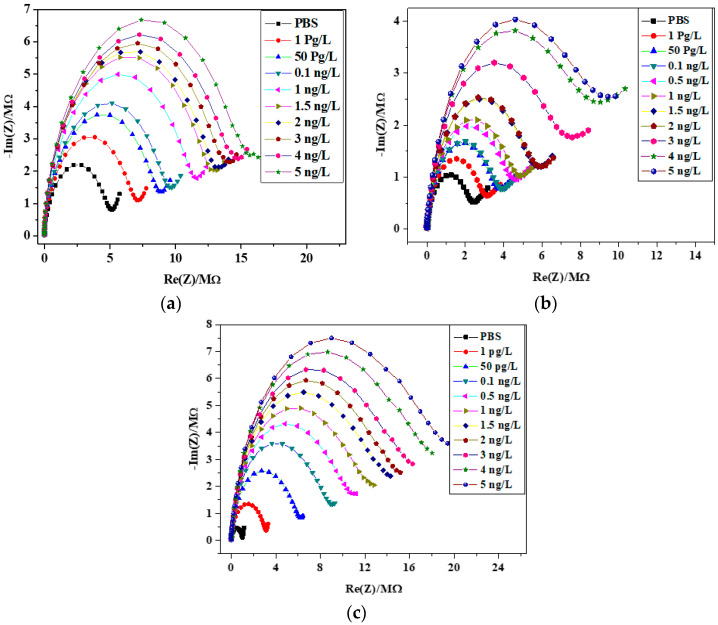
Nyquist diagrams of gold electrode/membrane interface (membrane 1, 2, 3) for different concentrations of tetracycline from 1 pg/L to 5 ng/L in a frequency range between 100 kHz and 6 Hz by applying a potential of −0.4 V and amplitude potential of 10 mV. (**a**) membrane 1; (**b**) membrane 2; (**c**) membrane 3.

**Figure 8 biosensors-13-00071-f008:**
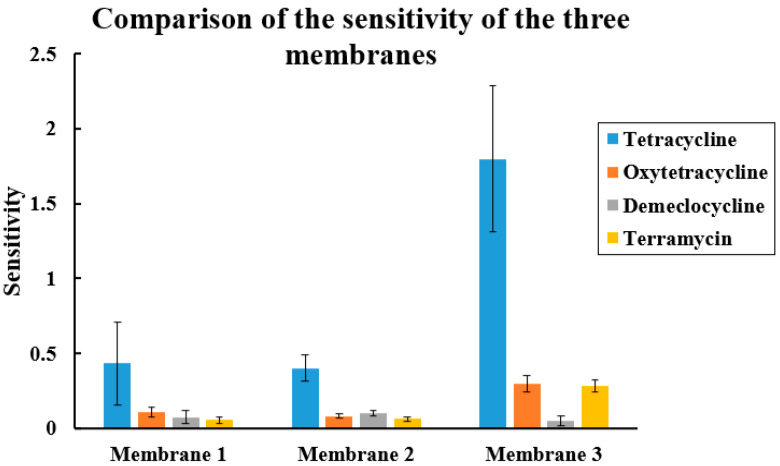
Sensitivities of membrane 1, membrane 2, membrane 3 sensors for tetracycline (blue) (concentrations ranging from 1 pg/L to 5 ng/L). Sensitivities for oxytetracycline (orange), demeclocycline (grey), and terramycin (yellow).

**Table 1 biosensors-13-00071-t001:** Comparison of the analytical performance of the prepared tetracycline sensor with membrane 3 and published electrochemical tetracycline biosensors.

Technique	Electrode	Immobilizing Biomolecules	Analyte	Linear Range (mol/L)	LOD (mol/L)	Refs
Photoelectrochemical aptasensor	cerium (Ce) doped CdS modified graphene (G)/BiYWO6	TC aptamer	Drug	4.5 × 10^−10^−2.25 × 10^−6^	2.25 × 10^−11^	[47]
EIS	integrated bio micro-electromechanical system (Bio-MEMS) Au	Anti-TC Polyclonal antibody	Honey	2.25 × 10^−13–^2.25 × 10^−9^	2.7 × 10^−12^	[16]
EIS	interdigitated array microelectrodes (IDAMs)	TC aptamer	Milk	1 × 10^−10–^1 × 10^−3^	3 × 10^−9^	[55]
EIS	Au	TC aptamer	Milk	2.25 × 10^−8–^6.75 × 10^−6^	2.25 × 10^−8^	[56]
EIS	Au	TC aptamer	Milk	1.12 × 10^−8–^1.12 × 10^−5^	2.25 × 10^−9^	[57]
EIS	glassy carbon electrode (GCE) modified with graphene oxide nanosheets	TC aptamer	Tablet/serum	1 × 10^−13–^1 × 10^−5^	29 × 10^−15^	[58]
EIS	nano-porous silicon (PS)	TC aptamer	-	2.1 × 10^−9–^62.4 × 10^−9^	2 × 10^−9^	[59]
EIS	nanomaterial modified with pencil graphite electrode	TC aptamer	Milk	1 × 10^16^– 1 × 10^–6^	3 × 10^–17^	[60]
EIS	glassy carbon electrode	TC aptamer	Honey	1 × 10^−16^–1 × 10^−6^	3.7 × 10^−17^	[61]
EIS	carbon paste electrode (CPE)	TC aptamer	Drug/Milk/Honey/blood serum	1 × 10^−14–^1 × 10^−6^	3.8 × 10^−15^	[62]
EIS	interdigital array microelectrode (IDAM)	TC aptamer	Milk	1 × 10^−9–^ 1 × 10^−3^	1 × 10^−9^	[63]
differential pulse voltammetry (DPV)	Gold electrode	Anti-TC monoclonal antibody	Milk	1.8 × 10^−10^–2.25 × 10^−9^	7.22 × 10^−11^	[64]
Amperometric immunosensor	Screen-printed dual carbon electrodes (SPdCEs)	Polyclonal sheep anti-TC antibody	Milk	1.12 × 10^−12^–1.12 × 10^−7^	1.93 × 10^−9^	[65]
LSV	Gold electrode electropolymerization of PATP functionalized AuNPs	MIP	Honey	2.24 × 10^−13^–2.24 × 10^−8^	2.2 × 10^−16^	[53]
EIS	PVC liquid membrane	[*o*-COSAN]^−^/TC ion-pair complex	Honey	2.25 × 10^−15–^1.12 × 10^−11^	7.5 × 10^−16^	This work

**Table 2 biosensors-13-00071-t002:** Determination of TC in spiked honey samples through the standard addition method, using tetracycline sensor (membrane 3).

**Samples**	**Added** **pg/L**	**Found** **pg/L**	**Recovery** **(%)**
Honey 1	50	53.5	107 ± 6
	500	515	103 ± 2
Honey 2	50	51	102 ± 6
	500	500	100 ± 6

## Data Availability

The data that support the findings of this study are available from the corresponding author upon reasonable request.

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
