# Peer review of "Selective Antibody-Free Sensing Membranes for Picogram Tetracycline Detection"

_biosensors, 2022, doi:10.3390/bios13010071_

Round 1
Reviewer 1 Report
In this manuscript, authors reported an electrochemical sensor for sensitive antibiotic tetracycline detection. Based on the same group’s previous research, a θ-shaped molecule
[Co(C2B9H11)2] were used to form the ion pair complex [tetracycline-H] [Co(C2B9H11)2] for antibiotic tetracycline recognition. Single-walled carbon nanotubes were deposited on gold microelectrodes for improved conductivity of the electrochemical sensor. It is interesting for developing sensitive and selective antibody-free sensing approaches for antibiotic detection in the future. However, the manuscript has several points that need to be corrected before acceptance.
1. There are so many grammar/typo mistakes in the manuscript, please check the entire manuscript carefully. For instance, LINE 74, tuneable should be tunable; LINE 34, mean disadvantage should be main disadvantage.
2. Details should be given in the Materials and Methods for microelectrodes fabrication. What was the spinning speed for photoresist layer? Which etchant was used for photoresist etching?
3. The mechanism for selective antibiotic tetracycline detection should be discussed in detail. There is a molecular structure of cosane/tetracycline ion pair complex in Figure 2. Is there any possibility of hydrogen bonding, hydrophobic effect or pi stacking involved in the complex formation?
4. The limitations of the method reported in the current manuscript should be discussed in detail.
5. The figures were rough in the current manuscript. Y axis was missing in Figure 9. The font in Figure 5 and 7 were blur.
6. Please check the formats for figures and tables required in this journal.
Author Response
Reviewer 1
In this manuscript, authors reported an electrochemical sensor for sensitive antibiotic tetracycline detection. Based on the same group’s previous research, a θ-shaped molecule
[Co(C2B9H11)2] were used to form the ion pair complex [tetracycline-H] [Co(C2B9H11)2] for antibiotic tetracycline recognition. Single-walled carbon nanotubes were deposited on gold microelectrodes for improved conductivity of the electrochemical sensor. It is interesting for developing sensitive and selective antibody-free sensing approaches for antibiotic detection in the future. However, the manuscript has several points that need to be corrected before acceptance.
We thank the reviewer for his comments and are grateful for his judgment on our article.
- There are so many grammar/typo mistakes in the manuscript, please check the entire manuscript carefully. For instance, LINE 74, tuneable should be tunable; LINE 34, mean disadvantage should be main disadvantage.
We thank the reviewer for his comments, the entire manuscript was checked by a native English person.
- Details should be given in the Materials and Methods for microelectrodes fabrication. What was the spinning speed for photoresist layer? Which etchant was used for photoresist etching?
A first photoresist layer is then applied with a spinning speed of 3000 rpm and patterned…
Etching away of the exposed photoresist was performed with the developer OPD4262 from Fujifilm, the left…
- The mechanism for selective antibiotic tetracycline detection should be discussed in detail. There is a molecular structure of cosane/tetracycline ion pair complex in Figure 2. Is there any possibility of hydrogen bonding, hydrophobic effect or pi stacking involved in the complex formation?
This point was clearly explained in lines 380-396.
“The proposed mechanism is based on the exchange process that occurs at the membrane interface with the analyte and is consequence of the geometry and chemical composition of the cobaltabis( dicarbollide) [3,3'-Co( 1,2-C2B9H 11 )2]-, particularly the existence of B-H and Ccluster-H bonds that can generate hydrogen and dihydrogen bonds. We believe this is essential. ln their absence, common ion exchangers will not generate strong interactions neither with the ammonium cation nor with the plasticizer solvent. Therefore, they have high mobility in the membrane. This is not the case with [3,3'-Co( 1,2-C2B9H 11 )2]- which does make these strong non-bonding interactions and generates more reticulate and therefore more stability and a precise concentration of the analyte in the membrane. The cobaltabis(dicarbollide) has dimensions of 1.1 nm in length and 0.6 nm in width and, is surrounded by hydrogens, the mentioned B-H bonds, that have considerable hydride character but not enough to be unstable in protic solvents. This sufficient hydride character of the B-H groups enables it to interact strongly with H-N units. The non-bonding interactions B-H···H-N are weak, but if there are many, as is the case, they become a strong interaction. This leads to a very stable concentration of the target analyte within the membrane and allows for an adequate ion exchange thanks to the ion exchanger capacity of [3,3’-Co(1,2-C2B9H11)2]-, overall giving the appropriate stability and sensitivity.”
- The limitations of the method reported in the current manuscript should be discussed in detail.
The advantages and the limitations of the method reported in the current manuscripts are respectively reported in lines 363-372 and lines 397-401
- The figures were rough in the current manuscript. Y axis was missing in Figure 9. The font in Figure 5 and 7 were blur.
We thank the reviewer for his comments, Y axis was added in Figure 9, we tried to improve the quality of figures 5 and 7 as much as possible.
- Please check the formats for figures and tables required in this journal.
Formats of figures and tables are in agreement with the requirements of the journal.

Reviewer 2 Report
In this study, the authors developed a membrane capable of selectively detecting tetracycline without antibodies and obtained the detection signal through EIS. The developed sensor was applied to real honey samples to detect tetracycline. This study is considered suitable for publication in Biosensors. However, there are the following insufficiency, so a revision is required through the addition of explanations.
1. It is complicated because there are too many figures. It is necessary to collect figures that explain similar things and subcategorize them such as (a) and (b). Example) Modify existing Figure 1 and 2 to figure 1a and 1b.
2. The overall image quality of the figure is not good. Improve the clarity of the figure for the benefit of the reader. In particular, the legend in Figure 5 is difficult to read. The clarity of Figure 7 (FTIR) is not good. In addition, the legend of the Nyquist plot of Membrane 1 in Figure 8 is different in resolution and size from the rest.
3. 'Honey 2' in Table 1 and the number indicating the number of lines are overlapped.
4. In Figure 4, show a magnified view of the cyclic voltammetry obtained with the thiol immobilized electrode.
5. The description of the SEM image in Figure 6 is too short. How are the SWCNTs arranged, what are the small white rods (possibly SWCNTs) and black spots in the image?
6. At Figure 3, the microelectrodes used as working electrodes are divided into three sensing area. Why uses this format? Also, was the Nyquist plot obtained with the three membranes in Figure 8 obtained at the same time (with one electrode as in Figure 3)? If not, do you get them individually or sequentially? A detailed description of the detection method gives great information about the practicality of the develope sensor.
7. Looking at Figure 1, the structures of the four molecules are similar. How can the developed sensor be sensitive only to tetracycline among them?
Author Response
Reviewer 2
In this study, the authors developed a membrane capable of selectively detecting tetracycline without antibodies and obtained the detection signal through EIS. The developed sensor was applied to real honey samples to detect tetracycline. This study is considered suitable for publication in Biosensors. However, there are the following insufficiency, so a revision is required through the addition of explanations.
We thank the reviewer for the comments and are grateful for his judgment on our article.
- It is complicated because there are too many figures. It is necessary to collect figures that explain similar things and subcategorize them such as (a) and (b). Example) Modify existing Figure 1 and 2 to figure 1a and 1b.
We thank the reviewer for the comments, we modify gathered figures 1 and 2 to figure 1a and 1b as requested.
- The overall image quality of the figure is not good. Improve the clarity of the figure for the benefit of the reader. In particular, the legend in Figure 5 is difficult to read. The clarity of Figure 7 (FTIR) is not good. In addition, the legend of the Nyquist plot of Membrane 1 in Figure 8 is different in resolution and size from the rest.
We thank the reviewer for the comments, we tried our best to improve the quality of the figures as requested, mainly figures 5, 6 and 7 (new figures 4, 5 and 6).
- 'Honey 2' in Table 1 and the number indicating the number of lines are overlapped.
Table 1 was redrawn
- In Figure 4, show a magnified view of the cyclic voltammetry obtained with the thiol immobilized electrode.
We thank the reviewer for the comments, In Figure 4 (new Figure 3) a magnified view of the cyclic voltammetry obtained with the thiol immobilized electrode is shown as requested.
- The description of the SEM image in Figure 6 is too short. How are the SWCNTs arranged, what are the small white rods (possibly SWCNTs) and black spots in the image?
We thank the reviewer for his comments, effectively the immobilization of SWCNTs onto the microelectrode was confirmed by the small white rods, however, the black spots in the image some spots not covered by SWCNTs.
- At Figure 3, the microelectrodes used as working electrodes are divided into three sensing area. Why uses this format? Also, was the Nyquist plot obtained with the three membranes in Figure 8 obtained at the same time (with one electrode as in Figure 3)? If not, do you get them individually or sequentially? A detailed description of the detection method gives great information about the practicality of the developed sensor.
We thank the reviewer for the comments, the microelectrode used as a working electrode present three sensing areas in order to easily study the repeatability of our developed biosensor. The Nyquist plot obtained with the three membranes in figure 8 (new figure 7) are obtained individually. Each chip that contains three sensing areas was functionalized with thiol, SWCNTs, and membrane (1 or 2 or 3). The reported results are the mean of the three sensing areas. This point was reported in line 324.
- Looking at Figure 1, the structures of the four molecules are similar. How can the developed sensor be sensitive only to tetracycline among them?
We thank the reviewer for the comments. This point was discussed in lines 400-404.
The higher sensitivity for tetracycline compared to the similar molecules should be explained by a higher stability of the [o-COSAN]-/tetracycline ion pair complex, whereas the similar molecules present a primary amine group which is a prerequisit, except terramycin. Demeclocycline presents a chloride group and oxytetracycline a supplementary OH group which can limit the ion pair complex stability.

Round 2
Reviewer 1 Report
I have no further comments.
Reviewer 2 Report
The quality of this manuscript has been adequately improved and suitable for publication in Biosensors.